**PLOS | ONE**

# The association between role model presence and self-regulation in early adolescence: A cross-sectional study

Miharu Nakanishi[1]*, Syudo Yamasaki[2], Kaori Endo[2], Shuntaro Ando[3],
Yuko Morimoto[4], Shinya Fujikawa[3,5], Sho Kanata[6], Yusuke Takahashi[7], Toshi
A. Furukawa[8], Marcus Richards[9], Mariko Hiraiwa-Hasegawa[10], Kiyoto Kasai[3,11],
Atsushi Nishida[2]

1 Mental Health and Nursing Research Team, Tokyo Metropolitan Institute of Medical Science, Setagaya-ku, Tokyo, Japan, 2 Mental Health Promotion Project, Tokyo Metropolitan Institute of Medical Science, Setagaya-ku, Tokyo, Japan, 3 Department of Neuropsychiatry, Graduate School of Medicine, The University of Tokyo, Bunkyo-ku, Tokyo, Japan, 4 School of Advanced Sciences, the Graduate University for Advanced Studies, Hayama, Kanagawa, Japan, 5 Centre for Adolescent Health, Murdoch Children's Research Institute, Parkville, Victoria, Australia, 6 Department of Psychiatry, Teikyo University School of Medicine, Tokyo, Japan, 7 Hakubi Center for Advanced Research, Kyoto University, Yoshida-Honmachi, Sakyo-ku, Kyoto, Japan, 8 Department of Health Promotion and Human Behavior, Graduate School of Medicine/School of Public Health, Kyoto University, Yoshida Konoe-cho, Sakyo-ku, Kyoto, Japan, 9 MRC Unit for Lifelong Health and Ageing, University College London, London, United Kingdom, 10 School of Advanced Sciences, SOKENDAI (The Graduate University for Advanced Studies), Hayama, Kanagawa, Japan, 11 The International Research Center for Neurointelligence (WPI-IRCN) at The University of Tokyo Institutes for Advanced Study (UTIAS), Bunkyo-ku, Tokyo, Japan

☯ These authors contributed equally to this work.
* mnakanishi-tky@umin.ac.jp

**Data Availability Statement:** Data cannot be shared publicly because of ethical restrictions imposed by the institutional ethics committee, as data contain sensitive information on young

## Abstract

### Purpose

Self-regulation is the capacity to regulate attention, emotion, and behaviour to pursue long-term goals. The current study examined the associations between role model presence and self-regulation during early adolescence, controlling for hopefulness, using a large population-based data set from the Tokyo Teen Cohort study.

### Methods

Adolescents, aged 12 years, identified a role model using a single item on a paper questionnaire: 'Who is the person you most look up to?' Level of hopefulness was also assessed using a single question: 'To what extent do you feel hopeful about the future of your life?' Trained investigators evaluated self-regulation.

### Results

Of 2550 adolescents, 2279 (89.4%) identified a role model. After adjusting for level of hopefulness, identifying a role model was associated with higher levels of self-regulation in comparison to indications of no role model. Hopeful future expectations were also associated

adolescents. However, the data can be made available from the Tokyo Teen Cohort Data Operating Committee for all interested researchers upon requests sent to the committee. The initial contact for request should be addressed to Translational Research Planning and Management Office [renkei@igakuken.or.jp].

**Funding:** This work was supported by the Grant-in-Aid for Scientific Research on Innovative Areas from the Ministry of Education, Culture, Sports, Science, and Technology of Japan [grant numbers JP23118002, JP16H06398, JP16H06395]. This work was partly supported by UTokyo Center for Integrative Science of Human Behavior (CiSHuB) and the International Research Center for Neurointelligence (WPI-IRCN) at The University of Tokyo Institutes for Advanced Study (UTIAS). The funders had no role in study design, data collection and analysis, decision to publish, or preparation of the manuscript.

**Competing interests:** The authors have declared that no competing interests exist.

with higher self-regulation; however, the beta coefficient was smaller than role model presence in the multivariate linear regression analysis.

## Conclusions

Role model presence was significantly associated with higher self-regulation among early adolescents. Educational environments should focus on support for adolescents with no role models.

## Introduction

Adolescence is a phase of human growth and development that takes place between childhood and adulthood [1]. Psychological maladjustments during adolescence, such as low self-esteem [2], depression [3], and suicidal behaviour [4], have an adverse influence on health and well-being in adulthood. Recent developments in research have suggested a need for an increased emphasis on strength and resilience-based approaches to adolescent health and positive youth development [5, 6].

Self-regulation is a significant adjustment indicator of positive youth development [7, 8]. Self-regulation is defined as the capacity to override natural and automatic tendencies, desires, or behaviours to pursue long-term goals, even at the expense of short-term desires [9]. Higher levels of self-regulation earlier in life predict later academic achievement [10], improved health, and psychological well-being [11]. Aside from positive psychological development, self-regulation is also related to internalising (anxiety, depression, and somatic complaints) and externalising (aggression, conduct problems, and delinquency) psychopathologies [12]. However, self-regulation is positive when an individual engages in goal-directed behaviour [13]. High levels of self-regulation have been observed in female adolescents [10, 14] and are associated with early language skills [14], high maternal education levels [14], and hopeful expectations for the future [15]. However, the correlates and sources of self-regulation have been less frequently explored.

Role model presence is a form of positive psychological functioning, which can play a central role in shaping self-regulation during early adolescence. Role models are individuals whom adolescents admire. Since identity formation is a central focus during adolescence, individuals often look to adults to determine what is appropriate and acceptable behaviour, as well as to identify role models whom they would like to resemble [16]. Role model presence has been related to reduced health risk behaviours in adolescents [17–20]. In addition to role model presence, type of role model is associated with health-related behaviours [21]. Identifying an individual known to the adolescent as a role model is particularly associated with higher self-esteem [17].

Hopeful future expectations can facilitate positive development for adolescents [22]. Hope represents an individual's ability to formulate or envision goals that motivate the individual to plan a course of behaviour, and the ability to see a plan or process by which the goals might be accomplished [23]. Hopeful future expectations have been suggested to play an important role in the development of self-regulation across adolescence [15]. As hopefulness stems from a sense of purpose in life [24], role model presence may precede hopeful future expectations.

The associations between role model presence and self-regulation after adjusting for level of hopefulness have not been previously examined in early adolescence. Clarifying the association between role model presence and self-regulation will provide a better understanding of the processes involved in positive youth development.

The present study aimed to investigate the associations between role model presence and self-regulation, controlling for hopefulness, among early adolescents, using a large population-based cohort, the Tokyo Teen Cohort (TTC). We hypothesised that self-regulation would be higher among adolescents who could identify a role model, even after controlling for their level of hopeful future expectations.

## Materials and methods

### Design

The present study used a cross-sectional design, based on data from the TTC. The TTC is a population-based longitudinal cohort study investigating the health and development of adolescents in three metropolitan areas of Tokyo, Japan [25]. The TTC consists of a baseline survey of 10-year-old children, a two-year follow-up survey and supplementary information provided by respondents' families. The present study used data from the first follow-up survey, when the respondents were 12 years old. Further details of the TTC are available in the literature [26, 27].

During the first follow-up survey, adolescents and their primary caregivers were asked to complete a self-administered, anonymous questionnaire. Trained investigators visited each participant's home twice between October 2012 and January 2015. After obtaining informed consent from both the adolescents and their caregivers during the first visit, participants were asked to complete the paper-based questionnaire. During the second visit, assessments were conducted using questionnaires and interviews. Questionnaires were completed anonymously and returned to the research team in closed envelopes.

All procedures involving human participants were performed in accordance with the ethical standards of the associated institutional research committees and in adherence to the 1964 Helsinki declaration and its later amendments, or comparable ethical standards. Informed consent was obtained from all adolescent respondents and their caregivers included in the study. All protocols were approved by the research ethics committees of the University of Tokyo Faculty of Medicine (approval no. 3150, 10057, and 10069), Tokyo Metropolitan Institute of Medical Science (approval no. 12–35), and the Graduate University for Advanced Studies (SOKENDAI) (approval no. 2012002).

### Participants

The baseline TTC survey included 3171 adolescents who were born between September 2002 and August 2004. Of the 3171 adolescents, 3007 (94.8%) participated in the first follow-up survey two years later.

### Measurements

The outcome variable in this study was self-regulation. Levels of self-regulation were measured by investigators. An investigator was asked to evaluate the self-control behaviours of each adolescent using a three-category response scale comparing their behaviours against those of a 'normal child' at age 12. Investigators completed a training course on self-regulation, including trial evaluations of at least 50 children. The self-regulation measure consisted of four items on behaviours related to attitude toward work, concentration, neatness in work, and daydreaming at home [28, 29]. The Japanese version of the instrument was developed as part of the current study. Total score was calculated using the sum of responses to the four items, with a range of 0 to 8. Higher scores indicated higher levels of self-regulation. In the present study, the internal

consistency of the self-regulation scale was moderate, with a Cronbach's alpha coefficient of 0.76.

The primary explanatory variable in this study was role model presence. Role model presence/absence was assessed by asking the participants a single open-ended question: 'Who is the person you most look up to?' Persons identified as role models were categorised as either individuals known to the adolescent (familial and non-familial), or figures primarily known through the media. Sub-categories referring to type of media figure were developed by a research panel, which included a registered nurse and clinical psychologist [21], based on the occupation groups for career aspirations identified in the Longitudinal Study of Australian Children [30]. Classification of each identified role model was performed by the research team (KE, in consultation with MN and SY). The results were compared, and differences were discussed until an agreement was reached regarding the classification of the identified role model.

The primary covariate in this study was hopeful future expectations. Level of hopefulness was assessed using a single question: 'To what extent do you feel hopeful about the future of your life?' Participants were asked to respond on a visual analogue scale from 0 (*not at all*) to 10 (*extremely*).

## Statistical analysis

The frequency of identification of each type of role model was calculated. Based on their responses, the adolescents were divided into four groups: (1) those identifying a familial individual; (2) those identifying a non-familial individual; (3) those identifying a media figure role model; and (4) those identifying no role model.

Level of hopefulness, according to role model presence and type of role model identified, were compared using a one-way analysis of variance (ANOVA). A Bonferonni correction was used for multiple pairwise tests.

Levels of self-regulation according to role model presence and type of role model identified were also compared using a one-way ANOVA. Since gender differences in self-regulation have been reported [14, 15], the total score differences for self-regulation between males and females were compared using a Student's *t* test. A pairwise correlation coefficient was calculated between hopeful future expectations and self-regulation.

Finally, a multiple linear regression analysis was conducted, using self-regulation as a dependent variable, and role model presence and type of role model as independent variables. Adolescent age and gender were included in the model as covariates. Hopeful future expectations score was then added to the multivariate model. In the regression analyses, full information maximum likelihood was used for estimating missing data [31].

All statistical analyses were conducted using Stata 15.1 (StataCorp, Texas) for Windows. The two-tailed significance level was set at .05.

## Results

Of the 3007 participants, 2550 with complete information regarding self-regulation were included in analyses. The mean age of the 2550 adolescents included in the analysis ($M = 12.1$ years) was significantly lower than that of the 457 excluded from the analysis [$M = 13.3$; $t(610.2) = 9.50$, $p < .001$]. Of the 2550 adolescent participants included in analyses, 1352 (53.0%) were male. The proportion of males was not significantly different from those excluded from the analysis [$\chi^2(1) = 0.04$, $p = .843$].

Of the 2550 adolescents, 2279 identified a role model. The types of role models are shown in Table 1. In the 2279 responses, participants identified role models who were family

**Table 1. Role model presence and type (N = 2550).**

| | N (%) | | |
|---|---|---|---|
| | Total | Male | Female |
| Total | 2550 (100.0) | 1352 (100.0) | 1198 (100.0) |
| Identified a role model | 2279 (89.4) | 1177 (87.1) | 1102 (92.0) |
| Individual known to the adolescent | 1518 (59.5) | 669 (49.5) | 849 (70.9) |
| Familial | 1114 (43.7) | 482 (35.7) | 632 (52.8) |
| Non-familial | 404 (15.8) | 187 (13.8) | 217 (18.1) |
| Figure primarily known through media | 761 (29.8) | 508 (37.6) | 253 (21.1) |
| No one identified as a role model | 154 (6.0) | 105 (7.8) | 49 (4.1) |
| Non-response to the question | 117 (4.6) | 70 (5.2) | 47 (3.9) |

members ($N$ = 1114, 43.7%), non-familial known individuals ($N$ = 404, 15.8%), and individuals known primarily through media ($N$ = 761, 29.8%). Of the 2550 adolescents, 154 (6.0%) answered 'no one' when asked to identify a role model. No response to the question was given by 117 (4.6%) of the participants. Significantly more male adolescents identified a figure known through the media when compared to females [$\chi^2(1)$ = 87.84, $p < .001$].

Of those who reported family members as role models ($N$ = 1114), 976 (87.6%) identified their parents. Non-familial known individuals, indicated as role models by 404 of the adolescents, included friends ($N$ = 184, 45.5%) and teachers ($N$ = 183, 45.3%).

A qualitative review of responses identifying figures known via media as role models ($N$ = 761) yielded 14 sub-categories based on occupational groups. Types of figure role models are presented in Table 2. Males were more likely to identify a sports player as a role model ($N$ = 208, 40.9%), whereas females were more likely to identify a historical figure who was devoted to humanity ($N$ = 68, 26.9%) or an artist ($N$ = 51, 20.2%).

Of the 2550 adolescents, 2539 responded to the question regarding hopefulness ($M$ = 7.3, $SD$ = 2.1). Level of hopefulness of future expectations according to role model is summarised in Table 3. Hopefulness was significantly higher in adolescents who identified a role model than those who identified no role model.

**Table 2. Types of figure role models among adolescents (N = 761).**

| | Common responses | N (%) | | |
|---|---|---|---|---|
| | | Total | Male | Female |
| Total | | 761 (100.0) | 508 (100.0) | 253 (100.0) |
| Sport player | Soccer player (Mr. Lionel Messi), baseball player | 236 (31.0) | 208 (40.9) | 28 (11.1) |
| Historical person devoted to humanity | Roman Catholic nun (Mother Teresa), Baptist minister (Dr. Martin Luther King, Jr.) | 161 (21.2) | 93 (18.3) | 68 (26.9) |
| Artist | Entrepreneur (Mr. Walt Disney), writer | 82 (10.8) | 31 (6.1) | 51 (20.2) |
| Performer | Actress (Ms. Miyako Yoshida, dancer), actor | 59 (7.8) | 22 (4.3) | 37 (14.6) |
| Technician | Engineer (Mr. Thomas Edison), Mechanic | 47 (6.2) | 41 (8.1) | 6 (2.4) |
| Self-Marketer | YouTube personality (Mr. Hajime Syacho) | 45 (5.9) | 26 (5.1) | 19 (7.5) |
| Science professional | Researcher (Mr. Albert Einstein) | 29 (3.8) | 21 (4.1) | 8 (3.2) |
| Political theorist, activist | Head of imperial family (Mr. Akihito), Prime Minister | 16 (2.1) | 11 (2.2) | 5 (2.0) |
| Explorer | Astronaut (Mr. Koichi Wakata) | 13 (1.7) | 10 (2.0) | 3 (1.2) |
| Doctor | Physician | 9 (1.2) | 7 (1.4) | 2 (0.8) |
| Nurse or caregiving professional | Nurse (Ms. Florence Nightingale), childcare worker | 6 (0.8) | 0 (0.0) | 6 (2.4) |
| Business professional or manager | Business magnate (Mr. Bill Gates) | 6 (0.8) | 6 (1.2) | 0 (0.0) |
| Character in fiction | Comic protagonist (Mr. Naruto Uzumaki) | 38 (5.0) | 22 (4.3) | 16 (6.3) |
| Others not specified | | 14 (1.8) | 10 (2.0) | 4 (1.6) |

**Table 3. Level of hopefulness of future expectations according to role model presence and type.**

| Mean (SD) | Familial (N = 1109) | Non-familial (N = 404) | Media figure (N = 760) | No role model (N = 153) | Test statistic | p-value |
|---|---|---|---|---|---|---|
| Hopeful future expectation (0–10) | 7.6 (1.9)[a,b] | 7.0 (2.3)[a,c,d] | 7.4 (2.2)[c,e] | 6.5 (2.2)[b,d,e] | $F(3) = 16.41$ | < .001 |

SD: standard deviation.

Hopefulness of future expectations was assessed using a visual analogue scale from 0 to 10.

[a,b,c,d,e] significant difference ($p < .05$) between the same alphabet, with Bonferroni correction.

The 2550 adolescents were rated by investigators with a mean self-regulation score of 5.7 (SD = 1.2). Self-regulation was significantly lower in males than females [$t(2505.1) = 2.55$, $p = .011$]. A significant positive correlation between hopeful future expectations and self-regulation ($r = 0.07$, $p = .001$) was observed.

Level of self-regulation according to role model presence and type is presented in Table 4. Self-regulation was significantly higher in adolescents who identified a role model than those who identified no role model.

Results of a multivariate linear regression analysis are shown in Table 5. After adjusting for age and sex, the analysis showed that adolescents with familial, non-familial, or media figure role models demonstrated higher total scores for self-regulation than those with no role models. Additionally, female participants demonstrated higher levels of self-regulation when compared to male adolescents. When adding hopeful future expectations as an independent variable in the model, the total score rating of self-regulation was higher in adolescents with a higher level of hopefulness. The significant positive association between role model and self-regulation remained after accounting for hopeful future expectations.

## Discussion

The present study is the first to indicate positive associations between role model presence and self-regulation in early adolescence. Specifically, in cases where adolescents identified a role model, higher levels of self-regulation were observed. Hopeful future expectations were also associated with higher self-regulation; however, the association between role model presence and self-regulation remained significant even after adjusting for level of hopeful future expectations.

Family members were the most commonly reported role model. However, male adolescents were more likely than females to identify media figures as role models, rather than known individuals. This pattern of role model identification is consistent with that observed in the United States [18, 21]. The male adolescents' responses may have reflected representations of gender in media texts [32], since young people tend to identify more strongly with gender-matched role models [21, 33]. Since some media figure role model types have been associated

**Table 4. Level of self-regulation according to role model presence and type.**

| Mean (SD) | Familial (N = 1114) | Non-familial (N = 404) | Media figure (N = 761) | No role model (N = 154) | Test statistic | p-value |
|---|---|---|---|---|---|---|
| Self-regulation (0–8) | 5.7 (1.3)[a] | 5.6 (1.2)[b] | 5.7 (1.2)[c] | 5.4 (1.2)[a,b,c] | $F(3) = 3.05$ | .027 |

SD: standard deviation.

Self-regulation was assessed using the 4-item scale of self-control behaviours.

[a,b,c] significant difference ($p < .05$) between the same alphabet, Bonferroni correction.

**Table 5. Estimated coefficients (Beta) predicting level of self-regulation.**

| | Adjusted for age and sex | +Adjusted for hopefulness |
|---|---|---|
| | Beta (95%CI) | Beta (95%CI) |
| Role model, reference = no role model | | |
| Familial | 0.27 (0.06, 0.48)* | 0.23 (0.02, 0.44)* |
| Non-familial | 0.29 (0.06, 0.53)* | 0.28 (0.04, 0.51)* |
| Media figure role model | 0.31 (0.09, 0.52)* | 0.27 (0.06, 0.49)* |
| Hopeful future expectations (0–10) | – | 0.03 (0.01, 0.06)* |
| Age, month | 0.003 (-0.01, 0.02) | 0.002 (-0.01, 0.02) |
| Sex, female | 0.12 (0.02, 0.22)* | 0.12 (0.02, 0.22)* |

$N$ = 2550; full information maximum likelihood estimation was used.

CI: confidence interval.

* Significant at $p < .05$.

Hopefulness of future expectations was assessed using a visual analogue scale from 0 to 10.

Self-regulation was assessed using the 4-item scale of self-control behaviours.

with increased health risk behaviours [21], further examination of the potentially negative effects of role models on health-related behaviours and self-regulation in adolescents is required.

Adolescents who have identified a role model have higher levels of self-regulation, as well as higher levels of hopeful future expectations. Both role model presence and hopeful future expectations were positively related to self-regulation in the final regression model. Therefore, role models may provide guidance in the formation of future expectations, contributing to higher self-regulation in adolescents in turn. The results of our study indicate that both media figure role models and known role models, whether familial or non-familial individuals, are positively associated with self-regulation. This finding emphasises the importance of the presence of role models external to adolescents' living environments, in addition to known individuals. In future, to enhance self-regulation, educational environments should focus on providing support for adolescents with no role models. In this study, some of the adolescents who did not respond to the role model question may have reacted differently if they were asked to identify a role model in an interview setting. In some cases, lack of response to the open question could have reflected a hesitation to identify 'no one' as a role model or being unable to think of a role model. This implies that there is a need for psychological support among these adolescents.

The greater self-regulation observed among female participants is consistent with the findings of previous, similar studies [10, 14]. Greater self-regulation in females may arise from cultural stereotypes, based on which females are more likely to mentally and behaviourally meet the needs of others [34]. Other factors relating to self-regulation among female adolescents have also been suggested, including the timing of rapid gains in the ability to regulate behaviour, early language skills, maternal education levels [14], and positive expectations for the future [15]. Our study contributes to the literature by examining the intrinsic factors related to self-regulation at an earlier developmental stage of adolescence.

Some limitations of the present study should be considered. This study utilised cross-sectional survey data which did not allow for the assessment of the stability of role model presence and type of role model. This may be problematic, as adolescents' future expectations can change with age [35]. In the future, longitudinal changes in role model identification, future expectations, and self-regulation should be investigated as part of the developmental process.

Although the Japanese version of the instrument for measuring self-regulation was developed based on well-established measures, further examination may be warranted to establish its validity.

Despite the limitations, the present study is the first to indicate that a positive association exists between role model presence and self-regulation among early adolescents. Our results suggest that identifying a role model be beneficial to the development of high self-regulation among early adolescents. Furthermore, there is a stronger association between role model presence and self-regulation than between hopeful future expectations and self-regulation. In the future, educational environments should focus on providing support for adolescents with no role models. Longitudinal analyses will allow for further investigation of change in role model identification, future expectations, and self-regulation across the course of development.

## Supporting information

**S1 File. A questionnaire developed as a part of this study.**
(PDF)

## Acknowledgments

We are thankful to all the adolescents who participated in this study.

## Author Contributions

**Conceptualization:** Miharu Nakanishi, Syudo Yamasaki, Kaori Endo, Shuntaro Ando, Yuko Morimoto, Shinya Fujikawa, Sho Kanata, Yusuke Takahashi, Toshi A. Furukawa, Marcus Richards, Mariko Hiraiwa-Hasegawa, Kiyoto Kasai, Atsushi Nishida.

**Data curation:** Miharu Nakanishi, Syudo Yamasaki, Kaori Endo, Shinya Fujikawa, Mariko Hiraiwa-Hasegawa, Kiyoto Kasai, Atsushi Nishida.

**Formal analysis:** Miharu Nakanishi, Syudo Yamasaki, Kaori Endo.

**Funding acquisition:** Shuntaro Ando, Mariko Hiraiwa-Hasegawa, Kiyoto Kasai, Atsushi Nishida.

**Investigation:** Syudo Yamasaki, Shuntaro Ando, Shinya Fujikawa, Kiyoto Kasai, Atsushi Nishida.

**Methodology:** Miharu Nakanishi, Syudo Yamasaki, Kaori Endo, Shuntaro Ando, Shinya Fujikawa, Sho Kanata, Yusuke Takahashi, Toshi A. Furukawa, Marcus Richards, Mariko Hiraiwa-Hasegawa, Kiyoto Kasai, Atsushi Nishida.

**Project administration:** Miharu Nakanishi, Syudo Yamasaki, Atsushi Nishida.

**Resources:** Shuntaro Ando, Yusuke Takahashi, Kiyoto Kasai, Atsushi Nishida.

**Software:** Syudo Yamasaki, Atsushi Nishida.

**Supervision:** Kaori Endo, Shuntaro Ando, Yuko Morimoto, Shinya Fujikawa, Sho Kanata, Yusuke Takahashi, Toshi A. Furukawa, Marcus Richards, Mariko Hiraiwa-Hasegawa, Kiyoto Kasai, Atsushi Nishida.

**Validation:** Shuntaro Ando, Yusuke Takahashi.

**Visualization:** Shuntaro Ando, Yusuke Takahashi.

**Writing – original draft:** Miharu Nakanishi.

**Writing – review & editing:** Syudo Yamasaki, Kaori Endo, Shuntaro Ando, Yuko Morimoto, Shinya Fujikawa, Sho Kanata, Yusuke Takahashi, Toshi A. Furukawa, Marcus Richards, Mariko Hiraiwa-Hasegawa, Kiyoto Kasai, Atsushi Nishida.

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
