## [Decision Letter · Decision Letter 0]

5 Aug 2019

PONE-D-19-16626

Role model presence as a shaper of early adolescents’ self-regulation: A population-based study

PLOS ONE

Dear Dr. Nakanishi,

Thank you for submitting your manuscript to PLOS ONE. After careful consideration, we feel that it has merit but does not fully meet PLOS ONE’s publication criteria as it currently stands. Therefore, we invite you to submit a revised version of the manuscript that addresses the points raised during the review process.

We would appreciate receiving your revised manuscript by Sep 19 2019 11:59PM. To enhance the reproducibility of your results, we recommend that if applicable you deposit your laboratory protocols in protocols.io, where a protocol can be assigned its own identifier (DOI) such that it can be cited independently in the future. For instructions see: http://journals.plos.org/plosone/s/submission-guidelines#loc-laboratory-protocols

We look forward to receiving your revised manuscript.

Kind regards,

Geilson Lima Santana, M.D., Ph.D.

Academic Editor

PLOS ONE

3. Thank you for including your ethics statement: "All procedures performed in studies involving human participants were in accordance with the ethical standards of the institutional research committees (Tokyo Metropolitan Institute of Medical Science, the University of Tokyo, and SOKENDAI) and with the 1964 Helsinki declaration and its later amendments or comparable ethical standards. Informed consent was obtained from both all adolescents and their caregivers included in the study."

a) Please amend your current ethics statement to confirm that your named institutional review board or ethics committee specifically approved this study.

Reviewers' comments:

Reviewer's Responses to Questions

**Comments to the Author**

1. Is the manuscript technically sound, and do the data support the conclusions?

Reviewer #1: Yes

2. Has the statistical analysis been performed appropriately and rigorously? 

Reviewer #1: Yes

3. Have the authors made all data underlying the findings in their manuscript fully available?

Reviewer #1: Yes

4. Is the manuscript presented in an intelligible fashion and written in standard English?

Reviewer #1: Yes

5. Review Comments to the Author

Reviewer #1: This is an interesting study that examined an association between role model presence and self-regulation using a large sample. Self-regulation is an important in human development process. Clarifying the relationship between role model presence and self-regulation appears to contribute on future education and the relevant intervention. I found the only two revision points and other two discretionary revision points. I listed them below.

Revision

1. Title, hypothesis, and discussion

The authors can clarify the primary aim of this study and how hopefulness was treated in the study. The title indicates that the primary aim was to test association between role model presence and self-regulation, although the two hypotheses mentioned hopefulness. Also, the discussion focused mostly on role model and self-regulation, rather than hopefulness. If the hopefulness was important for this study, the authors can add their assumption of the significant results for hopefulness in the section of Discussion.

2. Limitation

It is suggested that the authors mention the validity of scales/measures which assessed main variables in this study.

Discretionary revision

1. Title

While the study was part of well-deigned large-scale cohort study in Japan, the present study employed a cross-sectional design. This indicates that the study could not create evidence on direction between exposure and outcome. Therefore, the terms “shaper” would not be appropriate. This term problem may apply to the same or similar terms throughout the manuscript. In addition, STROBE guideline (e.g. von Elm et al, BMJ, 2007) recommends the authors to write the study design in the title. The authors can reconsider the title. An example below;

An association between role model presence and self-regulation in early adolescents: a cross-sectional study.

2. Statistical analysis

In the final analysis, it appears that the authors included the variable for hopefulness, like a control variable or a co-variate. If the authors have two clear hypotheses about testing two associations between role model presence and self-regulation and between role model presence and hopefulness, each hypothetical relationship can be separately tested in the different regression models, because each relationship may theoretically have the different potential confounders for which the authors should be adjusted. On the other hand, if the authors want to examine the association between three variables including role model presence, self-regulation and hopefulness, together, structural equation modelling (SEM) would be appropriate as the final modelling analysis.

6. PLOS authors have the option to publish the peer review history of their article (what does this mean?). If published, this will include your full peer review and any attached files.

Reviewer #1: No

---

## [Author Response · Author response to Decision Letter 0]

29 Aug 2019

Comments from Reviewer 1

General comment

This is an interesting study that examined an association between role model presence and self-regulation using a large sample. Self-regulation is an important in human development process. Clarifying the relationship between role model presence and self-regulation appears to contribute on future education and the relevant intervention. I found the only two revision points and other two discretionary revision points. I listed them below.

Response the general comment

Thank you for your comments. We have incorporated the suggested changes into the manuscript to the best of my ability.

Comment 1

1. Title, hypothesis, and discussion

The authors can clarify the primary aim of this study and how hopefulness was treated in the study. The title indicates that the primary aim was to test association between role model presence and self-regulation, although the two hypotheses mentioned hopefulness. Also, the discussion focused mostly on role model and self-regulation, rather than hopefulness. If the hopefulness was important for this study, the authors can add their assumption of the significant results for hopefulness in the section of Discussion.

Response to the comment 1

Thank you for your comment. As suggested by the comment 1 and 4, our primary aim was to test association between role model presence and self-regulation. Therefore we have modified abstract, Introduction, Materials and Methods, and Discussion to make the point clear.

Abstract, page 4 line 49-52: The current study examined the associations between role model presence and self-regulation during early adolescence, controlling for hopefulness, using a large population-based data set from the Tokyo Teen Cohort study.

Introduction, page 7 line 101-107: Hopeful future expectations have been suggested to play an important role in the development of self-regulation across adolescence [15]. As hopefulness stems from a sense of purpose in life [24], role model presence may precede hopeful future expectations. The associations between role model presence and self-regulation after adjusting for level of hopefulness have not been previously examined in early adolescence.

Introduction, page 8 line 109-116: The present study aimed to investigate the associations between role model presence and self-regulation, controlling for hopefulness, among adolescents, using a large population-based cohort, the Tokyo Teen Cohort (TTC). We hypothesised that self-regulation would be higher among adolescents who could identify a role model, even after controlling for their level of hopeful future expectations.

Materials and Methods, page 11 line 173-174: The primary covariate in this study was hopeful future expectations.

Discussion, page 19 line 262-268: The present study is the first to indicate positive associations between role model presence and self-regulation in early adolescence. Specifically, in case where adolescents identified a role model, higher levels of self-regulation were observed. Hopeful future expectations were also associated with higher self-regulation; however, the association between role model presence and self-regulation remained significant even after adjusting for level of hopeful future expectations.

Comment 2

2. Limitation

It is suggested that the authors mention the validity of scales/measures which assessed main variables in this study.

Response to the comment 2

Thank you for the suggestion. As suggested by the editor’s comment 2, we have explained that the Japanese version of instrument for self-regulation was developed as part of our study in page 10 line 157-158. Thus, we have added the mention regarding validity to limitations.

Page 21 line 306 - page 22 line 308: Although the Japanese version of instrument for measuring self-regulation was developed based on well-established measures, further examination may be warranted too establish its validity.

Comment 3

1. Title

While the study was part of well-deigned large-scale cohort study in Japan, the present study employed a cross-sectional design. This indicates that the study could not create evidence on direction between exposure and outcome. Therefore, the terms “shaper” would not be appropriate. This term problem may apply to the same or similar terms throughout the manuscript. In addition, STROBE guideline (e.g. von Elm et al, BMJ, 2007) recommends the authors to write the study design in the title. The authors can reconsider the title. An example below;

An association between role model presence and self-regulation in early adolescents: a cross-sectional study.

Response to the comment 3

Thank you for the suggestion. We have revised the title as suggested by the comment, and removed all the terms ‘shaper’ from the manuscript.

Comment 4

2. Statistical analysis

In the final analysis, it appears that the authors included the variable for hopefulness, like a control variable or a co-variate. If the authors have two clear hypotheses about testing two associations between role model presence and self-regulation and between role model presence and hopefulness, each hypothetical relationship can be separately tested in the different regression models, because each relationship may theoretically have the different potential confounders for which the authors should be adjusted. On the other hand, if the authors want to examine the association between three variables including role model presence, self-regulation and hopefulness, together, structural equation modelling (SEM) would be appropriate as the final modelling analysis.

Response to the comment 4

Thank you for the comment. As suggested by the comment 1, we have revised the manuscript to define hopefulness as a covariate.

We wish to thank the reviewers again for the valuable comments.

---

## [Editor Report · Decision Letter 1]

9 Sep 2019

The association between role model presence and self-regulation in early adolescence: A cross-sectional study

PONE-D-19-16626R1

Dear Dr. Nakanishi,

We are pleased to inform you that your manuscript has been judged scientifically suitable for publication and will be formally accepted for publication once it complies with all outstanding technical requirements.

With kind regards,

Geilson Lima Santana, M.D., Ph.D.

Academic Editor

PLOS ONE

---

## [Editor Report · Acceptance letter]

11 Sep 2019

PONE-D-19-16626R1 

The association between role model presence and self-regulation in early adolescence: A cross-sectional study 

Dear Dr. Nakanishi:

I am pleased to inform you that your manuscript has been deemed suitable for publication in PLOS ONE. Congratulations! Your manuscript is now with our production department. 

With kind regards,

on behalf of

Dr. Geilson Lima Santana 

Academic Editor

PLOS ONE